# Model-based Predictive Security Control for Discrete Switching Systems Under Deception Attacks*

1st Jianghan Xu
*College of Marine Electrical Engineering*
*Dalian Maritime University*
Dalian, China
xjh@dlmu.edu.cn

2nd Lili Li
*College of Marine Electrical Engineering*
*Dalian Maritime University*
Dalian, China
lilili@dlmu.edu.cn

3rd Mengjie Li
*College of Marine Electrical Engineering*
*Dalian Maritime University*
Dalian, China
1452429847@qq.com

4th Bin Lu
*College of Marine Electrical Engineering*
*Dalian Maritime University*
Dalian, China
adultlb@163.com

5th XiaoWei Zhao
*College of Marine Electrical Engineering*
*Dalian Maritime University*
Dalian, China
zhaoxw@dlmu.edu.cn

*Abstract*—This paper focuses on model prediction-based security control for discrete switching systems under deception attacks. An event-triggering mechanism based on the optimal prediction is designed to shorten the asynchronous time between the subsystem and the controller, which helps the system adjust when the prediction deviates and reduces the impact of the attack on the system performance. A new quadratic cost function is proposed to find a smoother and more stable control strategy; a subsystem switching rule based on the optimal prediction state is constructed to select the optimal subsystem switching and optimize the overall performance of the system; and a sufficient condition for asymptotic stabilization of the closed-loop system under the above event-triggering mechanism and optimal safety control strategy is given.

*Index Terms*—Safety control, Switched systems, Attack detection, Network attack, Predictive control

## I. INTRODUCTION

Networked switching systems describing multi-modalisation models have received increasing attention as the complexity of control objects in various types of real industrial systems continues to rise [1]. Due to the intertwining of control and network, the two main challenges are resource scheduling under communication constraints and attack prevention in open environments. To cope with the resource scheduling problem, the event-triggering mechanism has become an effective solution [2] [3] [4]. Also due to the openness of the network, the switching system communication network frequently suffers from malicious attacks, seriously interfering with the normal operation of the system [5].

This work is supported by the National Nature Science Foundation of China under Grant 62273068, and the Natural Science Foundation of Liaoning Province under Grant 2023-MS-120.

Model predictive control shows significant advantages in solving the problem of switching system security control under attack [6]. However, when the switching system is subjected to deception attacks, the controller modes are tampered with several times and are not consistent with the actual system modes, which can lead to a complex and difficult optimization process for model predictive control. In order to solve this problem, new index coefficients are introduced into the quadratic cost function, which can portray the dynamic behaviour of the system more accurately by considering the rate of increase and decrease of the Lyapunov function [7], [8] investigated mean residence time-based model predictive control for switched linear systems with time-varying delays in data transmission. [9] developed a model predictive control framework for the co-optimization of switching sequences and control inputs of a switched linear system under continuous residence time constraints with guaranteed recursive feasibility and stability. [10] investigated observer-based model predictive control for discrete switching systems under event-triggered mechanisms and denial-of-service attacks. So as to better reflect the stability requirements of the system in the optimization problem, and alleviate the problems caused by modal mismatch to a certain extent. However, since the switching rules in this paper rely on hysteresis state switching, it is still an urgent challenge to design a new quadratic cost function using model prediction methods to optimize the overall performance of the system and enhance the security for the complex problem of multiple tampering of the controller modes in discrete switching systems under deception attacks.

This article specifically addresses the optimal model prediction-based security control problem for discrete switch-

ing systems under deception attacks. In contrast to prior studies, four distinct contributions are outlined below: 1) Designing the event triggering mechanism based on optimal prediction, the system can dynamically adjust the triggering strategy when the prediction deviates based on the comparison result between the prediction error and the preset threshold. 2) Designing a new quadratic cost function allows the controller to trade-off control performance and switching cost, avoiding costly switching and switching that may seem beneficial in the short term but may damage system performance or stability in the long term, and thus finding a smoother and more stable control strategy that reduces system fluctuations caused by switching. 3) Designing system switching rules based on the optimal prediction state helps to select the optimal subsystem during switching, thus optimizing the overall performance of the system and improving the safety of the system. 4) The complex asynchronous switching behavior between the controller and the system is analyzed under the multiple designs mentioned above and stability criteria are given.

## II. PROBLEM FORMULATION

### A. System Description

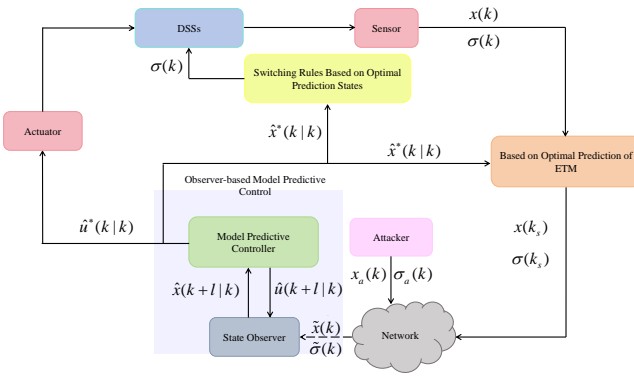

Fig. 1: Diagram of model predictive optimal security control strategy for discrete-time switched system under deception attacks.

As portrayed in Fig.1, the DSSs are described as

$$\begin{cases} x(k+1) = A_{\sigma(k)}x(k) + B_{\sigma(k)}u(k), \\ y(k+1) = C_{\sigma(k)}x(k), \end{cases} \quad (1)$$

where $x \subseteq \mathbb{R}^{n_x}$, $u \subseteq \mathbb{R}^{n_u}$, $y \subseteq \mathbb{R}^{n_y}$ are state, control input, regulated output. $A_\sigma$, $B_\sigma$, $C_\sigma$ are known appropriate matrices. The system state $x(k)$ and switching signal $\sigma(k)$ are sent to the event triggering mechanism based on optimal prediction, which determines whether the sampling data and the subsystem switching signal satisfy the triggering conditions and then transmitted to the controller side through the network. When deception attack occurs, both $x(k_s)$ and $\sigma(k_s)$ are tampered with and subject to network-induced delay. If the attack occurs multiple times, the healthy subsystem switching signal $\sigma(k)$ can be tampered with multiple times, which can lead to complex asynchronous switching behavior. The observer measures the system state $\hat{x}(k)$ for transmission

to the model predictive controller, after the quadratic cost function to obtain the optimal control signal $\hat{u}^*(k|k)$ and the corresponding optimal system state $\hat{x}^*(k|k)$. The control inputs $\hat{u}^*(k|k)$ are transmitted to the actuator for use by the system, and $\hat{x}^*(k|k)$ is used for the design of ETM and switching rules.

### B. Network and Deception attack

Since data is transmitted to the controller over network, it is vulnerable to transmission delays and Deception attacks. Define $\tau_s$ as the transmission delay at the $s$-th triggering instant. $\underline{\tau} \le \tau_s \le \bar{\tau}$, $\bar{\tau} = \max_{s \in \mathbb{N}^+}\{\tau_s\}$, $\underline{\tau} = \min_{s \in \mathbb{N}^+}\{\tau_s\}$. Besides, the $\iota$-th attack interval is expressed as $\Upsilon_\iota^1 = [\bar{d}_\iota, \underline{d}_\iota)$ and sleeping interval is $\Upsilon_\iota^2 = [\underline{d}_\iota, \bar{d}_{\iota+1})$. $\underline{d}_\iota = \bar{d}_\iota + \ell_\iota^a$ with duration $\ell_\iota^a$, $\iota \in \mathbb{N}^+$. Attacker injects an attack signal $x_a(k)$ into data $x(k)$ transmitted in the network, altering it to $\tilde{x}(k) = x(k) + \mu_x(k)x_a(k)$. If $k \in \Upsilon_\iota^1$, $\mu_x(k) = 1$, otherwise $\mu_x(k) = 0$. For $\Upsilon_\iota^1$, the healthy switching signal $\sigma(k)$ in the sensor-to-controller channel is tampered with by an erroneous switching signal $\tilde{\sigma}_\iota(k)$. Without loss of generality, Deception attacks must meet the assumed limits on duration and frequency [11].

## III. DESIGN OF OPTIMAL SECURITY CONTROL STRATEGY UNDER DECEPTION ATTACKS

### A. Event-Triggering Mechanism based on Optimal Prediction

In Fig. 1, the ETM is designed as

$$k_{s+1} = \min\{k_{s+1}^1 k_{s+1}^2 k_{s+1}^3 k_{s+1}^4\} \quad (2)$$

where $k_{s+1}^1 = \min_{r \in \mathbb{N}^+}\{\delta_{s,r}|f(\delta_{s,r}) > 0\}$ is the error detection condition, $f(\delta_{s,r}) = e_x^T(\delta_{s,r})\Phi_{\sigma(\delta_{s,r})}^1 e_x(\delta_{s,r}) - \gamma_{\sigma(\delta_{s,r})}x^T(\delta_{s,r})\Phi_{\sigma(\delta_{s,r})}^2 x(\delta_{s,r})$, $e_x(\delta_{s,r}) = x(\delta_{s,r}) - x(k_s)$ is the error of current state $x(\delta_{s,r})$ and triggering state $x(k_s)$. $\delta_{s,r} = k_s + r$ is the $r$-th sampling instant after $s$-th triggering instant, weighted matrices $\Phi_{\sigma(\delta_{s,r})}^1 > 0$, $\Phi_{\sigma(\delta_{s,r})}^2 > 0$. and the threshold $0 < \gamma_{\sigma(\delta_{s,r})} < 1$. $k_{s+1}^2 = \min_{r \in \mathbb{N}^+}\{\delta_{s,r}|[\varepsilon(\delta_{s,r})]^T\varepsilon(\delta_{s,r}) > \theta\}$ is the optimal prediction parameter error detection condition, $\varepsilon(\delta_{s,r}) = x(\delta_{s,r}) - \hat{x}^*(\delta_{s,r})$ is the error of current state $x(\delta_{s,r})$ and optimal prediction state $\hat{x}^*(\delta_{s,r})$. $\theta > 0$ is the given threshold value. $k_{s+1}^3 = \min_{r \in \mathbb{N}^+}\{\delta_{s,r}|\sigma(\delta_{s,r}) \neq \sigma(\delta_{s,r-1})\}$ is the modality matching condition, when modalities are distinct at adjacent sampling instants, an event triggering is arranged at $\delta_{s,r}$. $k_{s+1}^4 = k_s + \underline{\ell}$ is the attack parameter condition, $\underline{\ell}$ is the lower bound of the attack sleeping time. By introducing a lower limit on the duration of sleeping to limit the upper limit on the instant of two triggerings, it is guaranteed that at least one triggering will exist during the attack sleep to achieve synchronisation of the subsystem and controller modes.

For a detailed timing analysis, signal holding interval $\mathfrak{I}_s = [k_s + \tau_s, k_{s+1} + \tau_{s+1})$ is split into $\mathfrak{I}_s = \bigcup_{r=0}^{\bar{r}_s}\mathfrak{I}_{s,r}$ where

$$\mathfrak{I}_{s,r} = \begin{cases} [\delta_{s,r} + \tau_s, \delta_{s,r+1} + \tau_s), r = 0, \cdots, \bar{r}_s - 1, \\ [\delta_{s,\bar{r}_s} + \tau_s, k_{s+1} + \tau_{s+1}), r = \bar{r}_s, \end{cases}$$

with $\bar{r}_s = \min_{r \in \mathbb{N}^+}\{r|\delta_{s,r+1} + \tau_s \ge k_{s+1} + \tau_{s+1}\}$. Then, a piecewise time-delay function $\eta(k) = k - \delta_{s,r}$ satisfying $\underline{\eta} = \underline{\tau} \le \eta(k) \le$

$\bar{\eta} = \bar{\tau} + 1$. Under the optimal prediction-based event triggering mechanism (2) with deception attacks, the system switching signal can be described as $\tilde{\sigma}(k) = \sigma(k - \eta(k)) + \mu_\sigma(k)\sigma_a(k)$, $k \in \mathfrak{I}_{s,r}$, system state can be described as $\tilde{x}(k) = x(k - \eta(k)) - e(k) + \mu_x(k)x_a(k)$, $k \in \mathfrak{I}_{s,r}$.

### B. Optimal Security Control based on Model Prediction

Based on the received $\tilde{x}(k)$ and $\tilde{\sigma}(k)$, the full-dimensional state observer is constructed as

$$\hat{x}(k + 1) = A_{\tilde{\sigma}(k)}\hat{x}(k) + B_{\tilde{\sigma}(k)}\hat{u}(k) + L_{\tilde{\sigma}(k)}C_{\tilde{\sigma}(k)}[\tilde{x}(k) - \hat{x}(k)]k \in \mathfrak{I}_{s,r} \tag{3}$$

where $\hat{x}(k) \in \mathbb{R}^{n_x}$ is the state vector of the observer, $\hat{u}(k) \in \mathbb{R}^{n_u}$ is the observer control input vector, $L_{\phi(k)}$ is the observer gain matrix of appropriate dimensions to be determined, $\tilde{\sigma} = \tilde{\sigma}(k) \in \mathbb{P}$ calculated by the following hysteresis state switching rule

$$\tilde{\sigma}(k) = \begin{cases} \phi_1, & \text{if } \tilde{\sigma}(k-1) = \phi_1 \text{and } \hat{x}(k) \in \Omega_{i\phi_1} \\ \arg\min_{\phi_2 \in \mathbb{P}\backslash\phi_1}\{\Omega_{i\phi_2}\}, & \text{if } \tilde{\sigma}(k-1) = \phi_1 \text{and } \hat{x}(k) \in \Omega_{i\phi_1 i\phi_2}, \end{cases} \tag{4}$$

where $\Omega_{i\phi_1} = \{\hat{x}(k)|\hat{x}^T(k)(\mathscr{P}_{i\phi_1} - \mathscr{P}_{i\phi_2} + \mathcal{N}_{i\phi_1,i\phi_2})\hat{x}(k) < 0, \phi_2 \in \mathbb{P}\backslash\phi_1\}$, $\Omega_{i\phi_1,i\phi_2} = \{\hat{x}(k)|\hat{x}^T(k)(\mathscr{P}_{i\phi_1} - \mathscr{P}_{i\phi_2} + \mathcal{N}_{i\phi_1,i\phi_2})\hat{x}(k) = 0, \phi_2 \in \mathbb{P}\backslash\phi_1\}$ with initial value $\tilde{\sigma}(k_0) = \sigma(k_0)$, $k \in \mathfrak{I}_{s,r}$. $\mathcal{N}_{i\phi_1,i\phi_2}$ is an indeterminate matrix, $\Omega_{i\phi_1,i\phi_2}$ is boundary of $\Omega_{i\phi_1}$ and $\bigcup_{i=1}^{\bar{p}}\Omega_{i\phi_1} \in \mathbb{R}^{n_x}$.

Constructing a state observer-based prediction model from 3

$$\hat{x}(k + l + 1|k) = A_{\tilde{\sigma}(k)}\hat{x}(k + l|k) + B_{\tilde{\sigma}(k)}\hat{u}(k + l|k) + L_{\tilde{\sigma}(k)}C_{\tilde{\sigma}(k)}[\tilde{x}(k + l) - \hat{x}(k + l|k)] \tag{5}$$

where $\hat{x}(k + l|k)$ is the predicted state at instant $k + 1$ based on instant $k$, $\hat{u}(k + l|k)$ is the control input at instant $k + 1$ based on instant $k$, designed as

$$\hat{u}(k + l|k) = K_{\tilde{\sigma}(k)}\hat{x}(k + l|k), k \in \mathfrak{I}_{s,r} \tag{6}$$

Based on (5)(6), the model predictive control optimises the control input sequence $U(k) = \{\hat{u}(k|k), \hat{u}(k + 1|k), \cdots\}$ by minimising the quadratic cost function $\mathcal{G}_\infty(k)$ in the infinite horizon.

$$\min_{\hat{u}(k+l|k), l\geq 0} \mathcal{G}_\infty$$
$$\text{s.t.} \quad \mathbb{U} = \{\hat{u}(k + l|k) : \|\hat{u}(k + l|k)\| \leq \bar{u}\} \tag{7}$$

where

$$\mathcal{G}_\infty = \sum_{l=0}^{\infty} J_\infty(k + l|k)$$
$$J_\infty(k + l|k) = \|\hat{x}(k + l|k)\|_{S_{\hat{x}}}^2 + \|\hat{u}(k + l|k)\|_{S_{\hat{u}}}^2 + \|\hat{x}(k + l|k)\|_{\mathcal{N}_{i\phi_1,i\phi_2}}^2 \tag{8}$$

with $\|\hat{x}(k+l|k)\|_{S_{\hat{x}}}^2 = \hat{x}^T(k+l|k)S_{\hat{x}}\hat{x}(k+l|k)$, $\|\hat{u}(k+l|k)\|_{S_{\hat{u}}}^2 = \hat{u}^T(k+l|k)S_{\hat{u}}\hat{u}(k+l|k)$ and $\|\hat{x}(k+l|k)\|_{\mathcal{N}_{i\phi_1,i\phi_2}}^2 = \hat{x}^T(k+l|k)\mathcal{N}_{i\phi_1,i\phi_2}\hat{x}(k+l|k)$ denote state cost, input cost and switching cost, respectively. $S_{\hat{x}} > 0$ and $S_u > 0$ are symmetric weight matrices for the given state and control inputs, respectively. $\bar{u}$ is the size of the constraint.

Solve the optimisation problem (7) by applying only the first control input $\hat{u}^*(k|k) = \hat{u}(k|k)$ in the optimally obtained control signal sequence $U(k)$ to the system (1).

### C. Design of Switching rules based on Optimal Prediction States

The optimal control input $\hat{u}^*(k|k)$ is substituted into the state observer based prediction model (5) to obtain the optimal prediction state

$$\hat{x}^*(k + 1|k) = A_{\tilde{\sigma}(k)}\hat{x}^*(k|k) + B_{\tilde{\sigma}(k)}\hat{u}^*(k|k) + L_{\tilde{\sigma}(k)}C_{\tilde{\sigma}(k)}[\tilde{x}(k) - \hat{x}^*(k|k)]k \in \mathfrak{I}_{s,r} \tag{9}$$

The switching rule based on the optimal prediction state is designed as follows:

$$\sigma(k) = \begin{cases} j, & \sigma(k-1) = j, \hat{x}^*(k|k) \in \Omega_{j\phi} \\ \arg\min_{i\in\mathbb{P}\backslash j}\{\Omega_{i\phi}\}, & \sigma(k-1) = j, \hat{x}^*(k|k) \in \Omega_{j\phi,i\phi} \end{cases} \tag{10}$$

Its initial value $\sigma(\partial) = \arg\min_{j\in\mathbb{P}}\{[\hat{x}^*(\partial)]^T\mathscr{P}_{j\phi}\hat{x}^*(\partial)|\hat{x}^*(\partial) \in \Omega_{j\phi}\}$, $\partial \in [-\bar{\eta}, 0)$ and $\Omega_{j\phi} = \{\hat{x}^*(k|k)|[\hat{x}^*(k|k)]^T(\mathscr{P}_{j\phi} - \mathscr{P}_{i\phi} + \mathcal{N}_{j\phi,i\phi})\hat{x}^*(k|k) < 0i \in \mathbb{P}\backslash j\}$, $\Omega_{j\phi,i\phi} = \{\hat{x}^*(k|k)|[\hat{x}^*(k|k)]^T(\mathscr{P}_{j\phi} - \mathscr{P}_{i\phi} + \mathcal{N}_{j\phi,i\phi})\hat{x}^*(k|k) < 0i \in \mathbb{P}\backslash j\}$, $\mathcal{N}_{j\phi,i\phi}$ is an indeterminate matrix, $\Omega_{j\phi,i\phi}$ is boundary of $\Omega_{j\phi}$ and $\bigcup_{j=1}^{\bar{p}}\Omega_{j\phi} \in \mathbb{R}^{n_x}$.

### D. Closed-loop Systems and Control Objectives

Defining an error vector $\varepsilon(k) = x(k) - \hat{x}(k|k)$, the state vector is augmented as $X(k) = [\hat{x}^*(k|k)^T \ \varepsilon^T(k)]^T$, $k \in \mathfrak{I}_{s,r}$. Substituting equation (6) into the switching system (1) and the optimal prediction state (9), respectively, and combining the hysteresis state switching rule (4) and the switching rule based on the optimal prediction state (10), the closed-loop system can be represented as

$$X(k + 1) = \Lambda_\vartheta^1 X(k) + \Lambda_\vartheta^2 X(k - \eta(k)) + \Lambda_\vartheta^3 e_x(k) + \Lambda_\vartheta^4 x_a(k) \tag{11}$$

where $\vartheta \in \{\sigma\tilde{\sigma}|\sigma, \tilde{\sigma} \in \mathbb{P}\}$, $X(\partial) = \hat{X}(\partial), \partial \in [-\bar{\eta}, 0)$,

$$\Lambda_\vartheta^1 = \begin{bmatrix} A_{\tilde{\sigma}} + B_{\tilde{\sigma}}K_{\tilde{\sigma}} - L_{\tilde{\sigma}}C_{\tilde{\sigma}} & 0 \\ A_\sigma - A_{\tilde{\sigma}} + (B_\sigma - B_{\tilde{\sigma}})K_{\tilde{\sigma}} + L_{\tilde{\sigma}}C_{\tilde{\sigma}} & A_\sigma \end{bmatrix}$$

$$\Lambda_\vartheta^2 = \begin{bmatrix} L_{\tilde{\sigma}}C_{\tilde{\sigma}} & L_{\tilde{\sigma}}C_{\tilde{\sigma}} \\ -L_{\tilde{\sigma}}C_{\tilde{\sigma}} & -L_{\tilde{\sigma}}C_{\tilde{\sigma}} \end{bmatrix} \Lambda_\vartheta^3 = \begin{bmatrix} -L_{\tilde{\sigma}}C_{\tilde{\sigma}} \\ L_\phi C_\phi \end{bmatrix}$$

$$\Lambda_\vartheta^4 = \begin{bmatrix} \mu_x(k)L_{\tilde{\sigma}}C_{\tilde{\sigma}} \\ -\mu_x(k)L_{\tilde{\sigma}}C_{\tilde{\sigma}} \end{bmatrix}$$

Without loss of generality, it is assumed that the $j$-th and $i$-th subsystems are activated within the neighbouring switching intervals $[\kappa_{q-1}, \kappa_q)$ and $[\kappa_q, \kappa_{q+1})$. Under the influence of deception attacks and network delay, the change of modal signal $(\sigma, \tilde{\sigma})$ on any interval $[\kappa_q, \kappa_{q+1})$ can be classified into the seven cases in Table I, depending on the activation interval $\Upsilon_\iota^1$ of the $\iota$-th attack covering different subsystem or controller switching instants. For $k \in [\kappa_q, \kappa_{q+1})$, $\tilde{\kappa}_q$ is the controller switching instant after network delay for the subsystem switching instant $\kappa_q$.

**Case A.** For $k \in [\kappa_q, \kappa_{q+1})$, $\iota$-th attack activation interval $\Upsilon_\iota^1$ override subsystem switching instant $\kappa_q$, $\tilde{d}_\iota < \kappa_q < d < \kappa_{q+1}$, $\iota \in \mathbb{N}^+$. As the deception attack tampers with the switching signal, it causes the subcontroller mode $\tilde{\sigma}$ to switch from $j$ to $\tilde{\sigma}_\iota$ at $\tilde{d}_\iota$. Under the switching rule (10) based on the optimal prediction state, $(\sigma, \phi)$ changes from $(j, \tilde{\sigma}_\iota)$ to $(i, \tilde{\sigma}_\iota)$ after the switching instant $\kappa_q$. Under the influence of the attack sleeping interval triggering condition in (2), $(\sigma, \phi)$

becomes synchronised mode $(i,i)$. Therefore, $(\sigma,\phi) = (i,\tilde{\sigma}_\iota)$ and $(\sigma,\phi) = (i,i)$ hold on the intervals $\kappa_q, d$ ) and $d$ , $\kappa_{q+1}$).

**Case B.** For $k \in [\kappa_q, \kappa_{q+1})$, $\iota$-th attack activation interval $\Upsilon^1_\iota$ override subsystem switching instant $\kappa_q$, $\tilde{d}_\iota < \kappa_q < d$ $< \tilde{d}_{\iota+1} < \cdots < \tilde{d}_{\iota+\iota_q} < d$ $< \kappa_{q+1}$ $\iota \in \mathbb{N}^+$. As the deception attack tampers with the switching signal, it causes the subcontroller mode $\tilde{\sigma}$ to switch from $j$ to $\tilde{\sigma}_\iota$ at $\tilde{d}_\iota$. Under the switching rule (10) based on the optimal prediction state, $(\sigma,\phi)$ changes from $(j,\tilde{\sigma}_\iota)$ to $(i,\tilde{\sigma}_\iota)$ after the switching instant $\kappa_q$. Under the influence of the attack sleeping interval triggering condition in (2), $(\sigma,\phi)$ becomes synchronised mode $(i,i)$. At the end of the attack, $(\sigma,\phi)$ is synchronised to $(i,i)$ by the event-triggering mechanism (2) after $d$ . Therefore, $(\sigma,\phi) = (j,\tilde{\sigma}_\iota)$ and $(\sigma,\phi) = (i,\tilde{\sigma}_\iota)$ hold on the intervals $\tilde{d}_\iota, \kappa_q$) and $\kappa_q, d$ ).

**Case C.** Attack interval $\Upsilon^1_\iota \subseteq [\kappa_q, \kappa_{q+1})$ override controller switching instant $\tilde{\kappa}_q$. Under the switching rule (10) based on the optimal prediction state, $(\sigma,\phi)$ changes from $(j,j)$ to $(i,j)$ after the switching instant $\kappa_q$. As the deception attack initiates and ends triggering a sub-controller switching, $(\sigma,\phi)$ undergoes a consecutive asynchronous switching from $(i,j)$ to $(i,\tilde{\sigma}_\iota)$, $(i,i)$ at switching moments $\tilde{d}_\iota$, $d$ . Thus, $(\sigma,\phi) = (i,j)$ on the interval $(\kappa_q,\tilde{d}_\iota)$ and $(\sigma,\phi) = (i,\tilde{\sigma}_\iota)$ and $(\sigma,\phi) = (i,i)$ hold on the intervals $(\tilde{d}_\iota, d$ ) and $(d$ , $\kappa_{q+1})$, respectively.

**Case D.** Attack interval $\Upsilon^1_\iota \subseteq [\kappa_q, \kappa_{q+1})$ override controller switching instant $\tilde{\kappa}_q$. $\kappa_q < \tilde{d}_\iota < d$ $< \cdots < \tilde{d}_{\iota+\iota_q} < d$ $< \kappa_{q+1}$, $\iota_q \in \mathbb{N}^+$. Under the switching rule (10) based on the optimal prediction state, $(\sigma,\phi)$ changes from $(j,j)$ to $(i,j)$ after the switching instant $\kappa_q$. Under the deception attack, the controller mode $\phi$ changes from $i$ to $\tilde{\sigma}_{\bar\iota}$ to $i$ after the sub-controller switching instants $\tilde{d}_{\bar\iota}$ and $d$ . Thus, $(\sigma,\phi) = (i,j)$ on the interval $(\kappa_q,\tilde{d}_\iota)$ and $(\sigma,\phi) = (i,\tilde{\sigma}_{\bar\iota})$ and $(\sigma,\phi) = (i,i)$ hold on the intervals $(\tilde{d}_{\bar\iota}, d$ ) and $(d$ , $\tilde{d}_{\bar\iota+1})$, respectively. $\bar\iota \in \{\iota, \iota+1, \cdots, \iota+\iota_q\}$, $\tilde{d}_{\iota+\iota_q+1} = \kappa_{q+1}$.

**Case E.** Attack interval $\Upsilon^1_\iota \subseteq [\kappa_q, \kappa_{q+1})$ override controller switching instant $\tilde{d}_{\bar\iota}$. $\bar\iota \in \{\iota, \iota+1, \cdots, \iota+\iota_q\}$, $\iota_q \in \mathbb{N}^+$. Under the influence of the hysteresis state switching rule (4), the controller mode $\phi$ changes from $j$ to $i$ after the sub-controller switching instant $\tilde{\kappa}_q$. Under the deception attack, the controller mode $\phi$ changes from $i$ to $\tilde{\sigma}_{\bar\iota}$ to $i$ after the sub-controller switching instants $\tilde{d}_{\bar\iota}$ and $d$ . Thus, $(\sigma,\phi) = (i,j)$ on the interval $\kappa_q,\tilde{\kappa}_q)$ and $(\sigma,\phi) = (i,\tilde{\sigma}_{\bar\iota})$ and $(\sigma,\phi) = (i,i)$ hold on the intervals $(\tilde{d}_{\bar\iota}, d$ ) and $(d$ , $\tilde{d}_{\bar\iota+1})$, respectively. $\bar\iota \in \{\iota, \iota+1, \cdots, \iota+\iota_q\}$, $\tilde{d}_{\iota+\iota_q+1} = \kappa_{q+1}$.

**Case F.** There is no controller switching instant $\tilde{d}_{\bar\iota}$ induced by the $\bar\iota$-th attack during the attack interval $\Upsilon^1_{\bar\iota} \subseteq [\kappa_q, \kappa_{q+1})$, $\bar\iota \in \{\iota, \iota+1, \cdots, \iota+\iota_q\}$, $\iota \in \mathbb{N}^+$. Under the influence of the hysteresis state switching rule (4), the controller mode $\phi$ changes from $j$ to $i$ after the sub-controller switching instant $\tilde{\kappa}_q$. Therefore, $(\sigma,\phi) = (i,j)$ and $(\sigma,\phi) = (i,i)$ hold on the intervals $(\kappa_q, \tilde{\kappa}_q)$ and $(\tilde{\kappa}_q, \kappa_{q+1})$ respectively.

**Case G.** For $k \in [\kappa_q, \kappa_{q+1})$, Attack interval $\Upsilon^1_\iota \subseteq [\kappa_q, \kappa_{q+1})$ override sub-system switching instant $\kappa_{q+1}$, $\tilde{\kappa}_q < \tilde{d}_\iota < \kappa_{q+1} < d$ ,

$\iota \in \mathbb{N}^+$. Under the influence of the hysteresis state switching rule (4), a synchronised switching from $(i,j)$ to $(i,i)$ occurs in $(\sigma,\phi)$ after the sub-controller switching instant $\tilde{\kappa}_q$. After the sub-controller switching instant $\tilde{d}_\iota$, due to the deception attack $\phi$ is tampered from $i$ to $\tilde{\sigma}_\iota$. Therefore, $(\sigma,\phi) = (i,j)$ and $(\sigma,\phi) = (i,i)$ hold on the intervals $(\kappa_q, \tilde{\kappa}_q)$ and $(\tilde{\kappa}_q, \tilde{d}_\iota)$, and $(\sigma,\phi) = (i,\tilde{\sigma}_\iota)$ hold on the interval $[\tilde{d}_\iota, \kappa_{q+1})$.

Through the analysis of Cases A-G, the optimal security control problem based on model predictions for system (1) is transformed into control objectives: (i) The model predictive control design (7)-(8) is transformed into an optimisation problem with a quadratic cost function that ensures the asymptotic stability of the closed-loop system (11). (ii) Incorporating constraint $\|u(k+l|k)\| \leq \bar{u}$ as a linear matrix inequality into an observer-based model predictive control scheme.

## IV. MAIN RESULTS

**Theorem 1:** Given positive parameters $\bar\eta$, $\eta$, $\gamma_\sigma < 1$, model prediction control weight matrix $S_{\hat{x}} > 0$, $S_{\hat{u}} > 0$ and $\varsigma_{\tilde\vartheta,\bar\vartheta} < 0$, minimising an upper bound $\beta$ on $\mathcal{G}_\infty(k)$ to ensure that the closed-loop system (11) is asymptotically stable, if forthcoming conditions are met:

(i) There exist matrices $P_\vartheta > 0$, $Q_\vartheta > 0$, $R_\vartheta > 0$, $\Phi^1_\sigma > 0$, $\Phi^2_\sigma > 0$ and $M^1_\vartheta$ with $(\vartheta,\sigma,\phi) \in \{(ii,i,i), (ij,i,j), (i\tilde{\sigma}_{\bar\iota},i,\tilde{\sigma}_{\bar\iota})\}$, $\bar\iota \in \{\iota, \iota+1, \cdots, \iota+\iota_q\}$ satisfying

$$\Psi_\vartheta < 0, \tag{12}$$

$$\begin{bmatrix} R^1_\vartheta & M^1_\vartheta \\ \clubsuit & R^1_\vartheta \end{bmatrix} > 0. \tag{13}$$

(ii) There exist indeterminate matrice of appropriate dimensions $N_{\bar\vartheta,\bar\vartheta}$ satisfying

$$\Omega_\vartheta < 0, \tag{14}$$

$$N_{\bar\vartheta_1,\bar\vartheta_1} + N_{\bar\vartheta_1,\bar\vartheta_2} \leq N_{\bar\vartheta_1,\bar\vartheta_2} \tag{15}$$

$$N_{\vartheta(\kappa^-_0),\vartheta(\kappa^-_{q+1})} \leq 0 \tag{16}$$

(iii) There exist scalar $\beta > 0$ satisfying

$$V_\vartheta(k|k) \leq \beta, \tag{17}$$

$$\begin{bmatrix} -\beta^{-1}\bar{u}^2 & K_\phi \hat{I} \\ \clubsuit & -P_\vartheta \end{bmatrix} > 0. \tag{18}$$

The matrix $\Psi_\vartheta = \{\varphi^{\varrho_1,\varrho_2}_\vartheta\}$, $(\varrho_1, \varrho_2 = 1, \cdots, 6)$ is composed of

sub-blocks:

$$\varphi_\vartheta^{1,1} = (\Lambda_\vartheta^1)^T (P_\vartheta + \tilde{R}_\vartheta)\Lambda_\vartheta^1 - \tilde{R}_\vartheta \Lambda_\vartheta^1 - (\Lambda_\vartheta^1)^T \tilde{R}_\vartheta - P_\vartheta + \tilde{R}_\vartheta + Q_\vartheta^0$$

$$- R_\vartheta^0 + \hat{I}^T S_{\hat{x}} \hat{I} + \hat{I}^T K_\phi^T S_{\hat{u}} K_\phi \hat{I} + \sum_{\upsilon=1}^{\bar{p}} \varsigma_{\tilde{\vartheta},\bar{\vartheta}} \left( P_{\tilde{\vartheta}} - P_{\bar{\vartheta}} + N_{\tilde{\vartheta},\bar{\vartheta}} \right)$$

$$+ N_{\tilde{\vartheta},\bar{\vartheta}}, \quad \hat{I} = \begin{bmatrix} 1 & 0 \end{bmatrix},$$

$$\varphi_\vartheta^{1,2} = (\Lambda_\vartheta^1)^T (P_\vartheta + \tilde{R}_\vartheta)\Lambda_\vartheta^2 - \tilde{R}_\vartheta \Lambda_\vartheta^2, \varphi_\vartheta^{1,3} = R_\vartheta^0,$$

$$\varphi_\vartheta^{1,5} = (\Lambda_\vartheta^1)^T (P_\vartheta + \tilde{R}_\vartheta)\Lambda_\vartheta^3 - \tilde{R}_\vartheta \Lambda_\vartheta^3,$$

$$\varphi_\vartheta^{1,6} = (\Lambda_\vartheta^1)^T (P_\vartheta + \tilde{R}_\vartheta)\Lambda_\vartheta^4 - \tilde{R}_\vartheta \Lambda_\vartheta^4,$$

$$\varphi_\vartheta^{2,2} = (\Lambda_\vartheta^2)^T (P_\vartheta + \tilde{R}_\vartheta)\Lambda_\vartheta^2 - 2R_\vartheta^1 + M_\vartheta^1 + (M_\vartheta^1)^T + \gamma_\sigma Z^T \Phi_\sigma^2 Z,$$

$$Z = \begin{bmatrix} 1 & 1 \end{bmatrix}, \varphi_\vartheta^{2,3} = R_\vartheta^1 - (M_\vartheta^1)^T, \varphi_\vartheta^{2,4} = R_\vartheta^1 - M_\vartheta^1,$$

$$\varphi_\vartheta^{2,5} = (\Lambda_\vartheta^2)^T (P_\vartheta + \tilde{R}_\vartheta)\Lambda_\vartheta^3, \varphi_\vartheta^{2,6} = (\Lambda_\vartheta^2)^T (P_\vartheta + \tilde{R}_\vartheta)\Lambda_\vartheta^4,$$

$$\varphi_\vartheta^{3,3} = -Q_\vartheta^0 + Q_\vartheta^1 - R_\vartheta^0 - R_\vartheta^1, \varphi_\vartheta^{3,4} = M_\vartheta^1, \varphi_\vartheta^{4,4} = -Q_\vartheta^1 - R_\vartheta^1,$$

$$\varphi_\vartheta^{5,5} = (\Lambda_\vartheta^3)^T (P_\vartheta + \tilde{R}_\vartheta)\Lambda_\vartheta^3 - \Phi_\vartheta^1, \varphi_\vartheta^{5,6} = (\Lambda_\vartheta^3)^T (P_\vartheta + \tilde{R}_\vartheta)\Lambda_\vartheta^4,$$

$$\varphi_\vartheta^{6,6} = (\Lambda_\vartheta^4)^T (P_\vartheta + \tilde{R}_\vartheta)\Lambda_\vartheta^4, \tilde{R}_\vartheta = \eta^2 R_\vartheta^0 + (\bar{\eta} - \underline{\eta})^2 R_\vartheta^1,$$

$$(\vartheta, \sigma, \phi) \in \{(i,i,i),(ij,i,j),(i\tilde{\sigma}_{\bar{\iota}},i,\tilde{\sigma}_{\bar{\iota}})\}, \upsilon \in \{i,j,\tilde{\sigma}_{\bar{\iota}}\}$$

$$(\tilde{\vartheta}, \bar{\vartheta}) \in \{(j\tilde{\sigma}_\iota, i\tilde{\sigma}_\iota),(j,ij),(i\tilde{\sigma}_{\bar{\iota}},i),(i,i\tilde{\sigma}_{\bar{\iota}}),(ij,i\tilde{\sigma}_\iota),(ij,i)\},$$

$$(\tilde{\vartheta}_1, \bar{\vartheta}_1) \subseteq (\tilde{\vartheta}, \bar{\vartheta}), (\bar{\vartheta}_1, \bar{\vartheta}_2) \subseteq (\tilde{\vartheta}, \bar{\vartheta}),$$

$$(i\phi_1, i\phi_2) \in \{(i\tilde{\sigma}_{\bar{\iota}},i),(i,i\tilde{\sigma}_{\bar{\iota}}),(ij,i\tilde{\sigma}_\iota),(ij,i)\} \subseteq (\tilde{\vartheta}, \bar{\vartheta}),$$

$$(j\phi_2, i\phi) \in \{(j\tilde{\sigma}_\iota, i\tilde{\sigma}_\iota),(j,ij)\} \subseteq (\tilde{\vartheta}, \bar{\vartheta}),$$

$$P_{\tilde{\vartheta}} = \begin{bmatrix} \mathcal{P}_{\tilde{\vartheta}} & 0 \\ 0 & 0 \end{bmatrix}, P_{\bar{\vartheta}} = \begin{bmatrix} \mathcal{P}_{\bar{\vartheta}} & 0 \\ 0 & 0 \end{bmatrix}, N_{\tilde{\vartheta},\bar{\vartheta}} = \begin{bmatrix} \mathcal{N}_{\tilde{\vartheta},\bar{\vartheta}} & 0 \\ 0 & 0 \end{bmatrix}.$$

The matrix $\Omega_\vartheta = \{\zeta_\vartheta^{\pi_1,\pi_2}\}, (\pi_1, \pi_2 = 1, \cdots, 4)$ is composed of sub-blocks:

$$\zeta_\vartheta^{1,1} = (\Lambda_\vartheta^1)^T N_{\tilde{\vartheta},\bar{\vartheta}} \Lambda_\vartheta^1 - N_{\tilde{\vartheta},\bar{\vartheta}}, \zeta_\vartheta^{1,2} = (\Lambda_\vartheta^1)^T N_{\tilde{\vartheta},\bar{\vartheta}} \Lambda_\vartheta^2,$$

$$\zeta_\vartheta^{1,3} = (\Lambda_\vartheta^1)^T N_{\tilde{\vartheta},\bar{\vartheta}} \Lambda_\vartheta^3, \zeta_\vartheta^{1,4} = (\Lambda_\vartheta^1)^T N_{\tilde{\vartheta},\bar{\vartheta}} \Lambda_\vartheta^4, \zeta_\vartheta^{2,2} = (\Lambda_\vartheta^2)^T N_{\tilde{\vartheta},\bar{\vartheta}} \Lambda_\vartheta^2,$$

$$\zeta_\vartheta^{2,3} = (\Lambda_\vartheta^2)^T N_{\tilde{\vartheta},\bar{\vartheta}} \Lambda_\vartheta^3, \zeta_\vartheta^{2,4} = (\Lambda_\vartheta^2)^T N_{\tilde{\vartheta},\bar{\vartheta}} \Lambda_\vartheta^4, \zeta_\vartheta^{3,3} = (\Lambda_\vartheta^3)^T N_{\tilde{\vartheta},\bar{\vartheta}} \Lambda_\vartheta^3,$$

$$\zeta_\vartheta^{3,4} = (\Lambda_\vartheta^3)^T N_{\tilde{\vartheta},\bar{\vartheta}} \Lambda_\vartheta^4, \zeta_\vartheta^{4,4} = (\Lambda_\vartheta^4)^T N_{\tilde{\vartheta},\bar{\vartheta}} \Lambda_\vartheta^4$$

**Proof.** To fulfil control objectives (i), first, the selection of the Lyapunov candidate function is made:

$$V_\vartheta(k) = V_{P_\vartheta}(k) + V_{Q_\vartheta}(k) + V_{R_\vartheta}(k) \tag{19}$$

where

$$V_{P_\vartheta}(k) = X^T(k) P_\vartheta X(k),$$

$$V_{Q_\vartheta}(k) = \sum_{\rho=k-\underline{\eta}}^{k-1} X^T(\rho) Q_\vartheta^0 X(\rho) + \sum_{\rho=k-\bar{\eta}}^{k-1-\underline{\eta}} X^T(\rho) Q_\vartheta^1 X(\rho),$$

$$V_{R_\vartheta}(k) = \underline{\eta} \sum_{\rho=-\underline{\eta}}^{-1} \sum_{w=k+\rho}^{k-1} \Gamma^T(w) R_\vartheta^0 \Gamma(w) + (\bar{\eta} - \underline{\eta})$$

$$\sum_{\rho=-\bar{\eta}}^{-1-\underline{\eta}} \sum_{w=k+\rho}^{k-1} \Gamma^T(w) R_\vartheta^1 \Gamma(w), \quad \Gamma(w) = X(w+1) - X(w)$$

Taking difference of (19) along the trajectory of closed-loop system (11) yields

$$\Delta V_{P_\vartheta}(k) = X^T(k+1) P_\vartheta X(k+1) - X^T(k) P_\vartheta X(k),$$

$$\Delta V_{Q_\vartheta}(k) = X^T(k) Q_\vartheta^0 \Gamma(k) - X^T(k-\underline{\eta}) Q_\vartheta^0 X(k-\underline{\eta})$$

$$+ X^T(k-\underline{\eta}) Q_\vartheta^1 X(k-\underline{\eta}) - X^T(k-\bar{\eta}) Q_\vartheta^1 X(k-\bar{\eta}),$$

$$\Delta V_{R_\vartheta}(k) = \underline{\eta}^2 \Gamma^T(k) R_\vartheta^0 \Gamma(k) - \underline{\eta} \sum_{w=k-\underline{\eta}}^{k-1} \Gamma^T(w) R_\vartheta^0 \Gamma(w)$$

$$+ (\bar{\eta} - \underline{\eta})^2 \Gamma^T(k) R_\vartheta^1 \Gamma(k) - (\bar{\eta} - \underline{\eta}) \sum_{w=k-\bar{\eta}}^{k-\underline{\eta}-1} \Gamma^T(w) R_\vartheta^1 \Gamma(w).$$

Applying Lemmas from [12] to the summation terms in $\Delta V_{R_\vartheta}(k)$, it follows from condition (13) that

$$- \underline{\eta} \sum_{w=k-\underline{\eta}}^{k-1} \Gamma^T(w) R_\vartheta^0 \Gamma(w) \leq - \left[ \sum_{w=k-\underline{\eta}}^{k-1} \Gamma^T(w) \right]^T R_\vartheta^0 \left[ \sum_{w=k-\underline{\eta}}^{k-1} \Gamma^T(w) \right]$$

$$\leq - X^T(k) R_\vartheta^0 X(k) + 2X^T(k) R_\vartheta^0 X(k-\underline{\eta}) - X^T(k-\underline{\eta}) R_\vartheta^0 X(k-\underline{\eta}) \tag{20}$$

$$- (\bar{\eta} - \underline{\eta}) \sum_{w=k-\bar{\eta}}^{k-\underline{\eta}-1} \Gamma^T(w) R_\vartheta^1 \Gamma(w)$$

$$\leq - X^T(k-\underline{\eta}) R_\vartheta^1 X(k-\underline{\eta}) + 2X^T(k-\underline{\eta})(R_\vartheta^1 - M_\vartheta^1) X(k-\eta(k))$$

$$+ 2X^T(k-\underline{\eta}) M_\vartheta^1 X(k-\bar{\eta}) + X^T(k-\eta(k))[-2R_\vartheta^1 + M_\vartheta^1$$

$$+ (M_\vartheta^1)^T] X(k-\eta(k)) + 2X^T(k-\eta(k))(-M_\vartheta^1 + R_\vartheta^1) X(k-\bar{\eta})$$

$$- X^T(k-\bar{\eta}) R_\vartheta^1 X(k-\bar{\eta}) \tag{21}$$

The error judgement function can be rewritten as

$$e_x^T(k) \Phi_\sigma^1 e_x(k) \leq \gamma_\sigma X^T(k-\eta(k)) \bar{\Phi}_\sigma^2 X(k-\eta(k)) \tag{22}$$

where $\bar{\Phi}_\sigma^2 = Z^T \Phi_\sigma^2 Z, Z = \begin{bmatrix} 1 & 1 \end{bmatrix}$.

Then, substituting $\varsigma_{\tilde{\vartheta},\bar{\vartheta}} < 0$ into (4) and (10) yields $\sum_{\upsilon=1}^{\bar{p}} \varsigma_{\tilde{\vartheta},\bar{\vartheta}} X^T(k)(P_{\tilde{\vartheta}} - P_{\bar{\vartheta}} + N_{\tilde{\vartheta},\bar{\vartheta}}) X(k) \geq 0$. Substituting them with (20)-(22) into $\Delta V_\vartheta(k)$, according to condition (12), that

$$\Delta V_\vartheta(k) \leq \Delta V_{P_\vartheta}(k) + \Delta V_{Q_\vartheta}(k) + \Delta V_{R_\vartheta}(k)$$

$$+ \gamma_\sigma X^T(k-\eta(k)) \bar{\Phi}_\sigma^2 X(k-\eta(k)) - e_x^T(k) \Phi_\sigma^1 e_x(k)$$

$$+ \hat{x}^T(k) S_{\hat{x}} \hat{x}(k) + \hat{u}^T(k) S_{\hat{u}} \hat{u}(k) + \hat{x}^T(k) \mathcal{N}_{i\phi_1,i\phi_2} \hat{x}(k)$$

$$+ \sum_{\upsilon=1}^{\bar{p}} \varsigma_{\tilde{\vartheta},\bar{\vartheta}} X^T(k)(P_{\tilde{\vartheta}} - P_{\bar{\vartheta}} + N_{\tilde{\vartheta},\bar{\vartheta}}) X(k) \tag{23}$$

$$\leq \bar{X}^T(k) \Psi_\vartheta \bar{X}(k) < 0$$

where $\bar{X}(k) = [X^T(k) X^T(k-\eta(k)) X^T(k-\underline{\eta}) X^T(k-\bar{\eta}) e_x^T(k) x_a^T(k)^T]^T$ $(\tilde{\vartheta}, \bar{\vartheta}) \in \{(j\tilde{\sigma}_\iota, i\tilde{\sigma}_\iota),(j,ij),(i\tilde{\sigma}_{\bar{\iota}},i),(i,i\tilde{\sigma}_{\bar{\iota}}), (ij,i\tilde{\sigma}_\iota),(ij,i)\}$ $(i\phi_1, i\phi_2) \in \{(i\tilde{\sigma}_{\bar{\iota}},i),(i,i\tilde{\sigma}_{\bar{\iota}}),(ij,i\tilde{\sigma}_\iota),(ij,i)\}$

Second, discussed (19) on interval $(\kappa_q, \kappa_{q+1})$.

**Case A.** On the interval $(\kappa_q, d_{\widetilde{\iota}})$, $(\sigma, \phi) = (i, \tilde{\sigma}_\iota)$. On the interval $(d_{\widetilde{\iota}}, \kappa_{q+1})$, $(\sigma, \phi) = (i, i)$. According to (23) yields

$$V_{ii}(\kappa_{q+1}^-) \leq V_{ii}(d_{\widetilde{\iota}}^+), \quad k \in (d_{\widetilde{\iota}}, \kappa_{q+1}) \tag{24}$$

For controller switching instant $d_{\widetilde{\iota}}$ affected by deception attack parameters, according to (4) yields

$$V_{ii}(d_{\widetilde{\iota}}^+) = V_{i\tilde{\sigma}_\iota}(d_{\widetilde{\iota}}^-) + X^T(d_{\widetilde{\iota}})N_{i\tilde{\sigma}_\iota,i}X(d_{\widetilde{\iota}}) \tag{25}$$

On the interval $[\kappa_q, d_{\widetilde{\iota}})$, according to (23) yields

$$V_{i\tilde{\sigma}_\iota}(d_{\widetilde{\iota}}^-) \leq V_{i\tilde{\sigma}_\iota}(\kappa_q^+), \quad k \in (\kappa_q, d_{\widetilde{\iota}}) \tag{26}$$

Combining (14) and (24)-(26), it can be deduced that

$$
\begin{aligned}
V_{ii}(\kappa_{q+1}^-) &\leq V_{ii}(d_{\widetilde{\iota}}^+) = V_{i\tilde{\sigma}_\iota}(d_{\widetilde{\iota}}^-) + X^T(d_{\widetilde{\iota}})N_{i\tilde{\sigma}_\iota,i}X(d_{\widetilde{\iota}}) \\
&\leq V_{i\tilde{\sigma}_\iota}(\kappa_q^+) + X^T(d_{\widetilde{\iota}})N_{i\tilde{\sigma}_\iota,i}X(d_{\widetilde{\iota}}) \\
&\leq V_{i\tilde{\sigma}_\iota}(\kappa_q^+) + X^T(\kappa_q)N_{i\tilde{\sigma}_\iota,i}X(\kappa_q)
\end{aligned} \tag{27}
$$

**Case B.** On the interval $(\kappa_q, d_{\widetilde{\iota}})$, $(\sigma, \phi) = (i, \tilde{\sigma}_\iota)$. On the interval $(d_{\widetilde{\iota}}, \tilde{d}_{\bar{\iota}+1})$, $(\sigma, \phi) = (i, i)$, $\bar{\iota} \in \{\iota, \iota+1, \cdots, \iota+\iota_q\}$, $\tilde{d}_{\iota+\iota_q+1} = \kappa_{q+1}$. On the interval $(\tilde{d}_{\bar{\iota}}, d_{\widetilde{\iota}})$, $(\sigma, \phi) = (i, \tilde{\sigma}_{\bar{\iota}})$, $\bar{\iota} \in \{\iota+1, \cdots, \iota+\iota_q\}$. According to (23) yields

$$V_{ii}(\kappa_{q+1}^-) \leq V_{ii}(d_{\widetilde{\iota+\iota_q}}^+), k \in [d_{\widetilde{\iota+\iota_q}}, \kappa_{q+1}) \tag{28}$$

For controller switching instant $d_{\widetilde{\iota+\iota_q}}$ affected by deception attack parameters, according to (4) yields

$$V_{ii}(d_{\widetilde{\iota+\iota_q}}^+) = V_{i\tilde{\sigma}_{\iota+\iota_q}}(d_{\widetilde{\iota+\iota_q}}^-) + X^T(d_{\widetilde{\iota+\iota_q}})N_{i\tilde{\sigma}_{\iota+\iota_q},i}X(d_{\widetilde{\iota+\iota_q}}) \tag{29}$$

On the interval $(\tilde{d}_{\iota+\iota_q}, d_{\widetilde{\iota+\iota_q}})$, according to (23) yields

$$V_{i\tilde{\sigma}_{\iota+\iota_q}}(d_{\widetilde{\iota+\iota_q}}^-) \leq V_{i\tilde{\sigma}_{\iota+\iota_q}}(\tilde{d}_{\iota+\iota_q}^+), k \in (\tilde{d}_{\iota+\iota_q}, d_{\widetilde{\iota+\iota_q}}) \tag{30}$$

For controller switching instant $\tilde{d}_{\iota+\iota_q}$ caused by deception attacks, we get

$$V_{i\tilde{\sigma}_{\iota+\iota_q}}(\tilde{d}_{\iota+\iota_q}^+) = V_{ii}(\tilde{d}_{\iota+\iota_q}^-) + X^T(\tilde{d}_{\iota+\iota_q})N_{i,i\tilde{\sigma}_{\iota+\iota_q}}X(\tilde{d}_{\iota+\iota_q}) \tag{31}$$

Combining (14)-(15) and (28)-(31), it can be deduced that

$$
\begin{aligned}
V_{ii}(\kappa_{q+1}^-) &\leq V_{ii}(d_{\widetilde{\iota+\iota_q}}^+) = V_{i\tilde{\sigma}_{\iota+\iota_q}}(d_{\widetilde{\iota+\iota_q}}^-) + X^T(d_{\widetilde{\iota+\iota_q}})N_{i\tilde{\sigma}_{\iota+\iota_q},i}X(d_{\widetilde{\iota+\iota_q}}) \\
&\leq V_{i\tilde{\sigma}_{\iota+\iota_q}}(\tilde{d}_{\iota+\iota_q}^+) + X^T(d_{\widetilde{\iota+\iota_q}})N_{i\tilde{\sigma}_{\iota+\iota_q},i}X(d_{\widetilde{\iota+\iota_q}}) \\
&= V_{ii}(\tilde{d}_{\iota+\iota_q}^-) + X^T(\tilde{d}_{\iota+\iota_q})N_{i,i\tilde{\sigma}_{\iota+\iota_q}}X(\tilde{d}_{\iota+\iota_q}) + X^T(d_{\widetilde{\iota+\iota_q}})N_{i\tilde{\sigma}_{\iota+\iota_q},i}X(d_{\widetilde{\iota+\iota_q}}) \\
&\leq V_{ii}(\tilde{d}_{\iota+\iota_q}^-) + X^T(\tilde{d}_{\iota+\iota_q})(N_{i,i\tilde{\sigma}_{\iota+\iota_q}} + N_{i\tilde{\sigma}_{\iota+\iota_q},i})X(\tilde{d}_{\iota+\iota_q}) \\
&\leq V_{ii}(\tilde{d}_{\iota+\iota_q}^-) + X^T(\tilde{d}_{\iota+\iota_q})N_{i,i}X(\tilde{d}_{\iota+\iota_q}) \leq V_{ii}(\tilde{d}_{\iota+\iota_q}^-)
\end{aligned} \tag{32}
$$

Repeating the above derivation process on the interval $k \in [\tilde{d}_{\iota+1}, \kappa_{q+1})$ yields

$$V_{ii}(\kappa_{q+1}^-) \leq V_{ii}(\tilde{d}_{\iota+\iota_q}^-) \leq \cdots \leq V_{ii}(\tilde{d}_{\iota+1}^-) \tag{33}$$

On the interval $[d_{\widetilde{\iota}}, \tilde{d}_{\iota+1})$, according to (23) yields

$$V_{ii}(\tilde{d}_{\iota+1}^-) \leq V_{ii}(d_{\widetilde{\iota}}^+), \quad k \in [d_{\widetilde{\iota}}, \tilde{d}_{\iota+1}) \tag{34}$$

For controller switching instant $d_{\widetilde{\iota}}$ affected by deception attack parameters, according to (4) yields

$$V_{ii}(d_{\widetilde{\iota}}^+) = V_{i\tilde{\sigma}_\iota}(d_{\widetilde{\iota}}^-) + X^T(d_{\widetilde{\iota}})N_{i\tilde{\sigma}_\iota,i}X(d_{\widetilde{\iota}}) \tag{35}$$

Combining (15)-(16) and (33)-(35), it can be deduced that

$$
\begin{aligned}
V_{ii}(\kappa_{q+1}^-) &\leq V_{ii}(\tilde{d}_{\iota+\iota_q}^-) \leq \cdots \leq V_{ii}(\tilde{d}_{\iota+1}^-) \leq V_{ii}(d_{\widetilde{\iota}}^+) \\
&= V_{i\tilde{\sigma}_\iota}(d_{\widetilde{\iota}}^-) + X^T(d_{\widetilde{\iota}})N_{i\tilde{\sigma}_\iota,i}X(d_{\widetilde{\iota}}) \\
&\leq V_{i\tilde{\sigma}_\iota}(\kappa_q^+) + X^T(\kappa_q)N_{i\tilde{\sigma}_\iota,i}X(\kappa_q)
\end{aligned} \tag{36}
$$

**Case C.** On the interval $[\kappa_q, \tilde{d}_\iota)$, $(\sigma, \phi) = (i, j)$. On the interval $[\tilde{d}_\iota, d_{\widetilde{\iota}})$, $(\sigma, \phi) = (i, \tilde{\sigma}_\iota)$. On the interval $[d_{\widetilde{\iota}}, \kappa_{q+1})$, $(\sigma, \phi) = (i, i)$. Analyses on interval $[d_{\widetilde{\iota}}, \kappa_{q+1}]$ are the same as in case A, according to (24)-(25) yields

$$V_{ii}(\kappa_{q+1}^-) \leq V_{ii}(d_{\widetilde{\iota}}^+) = V_{i\tilde{\sigma}_\iota}(d_{\widetilde{\iota}}^-) + X^T(d_{\widetilde{\iota}})N_{i\tilde{\sigma}_\iota,i}X(d_{\widetilde{\iota}}), k \in [d_{\widetilde{\iota}}, \kappa_{q+1}) \tag{37}$$

On the interval $[\tilde{d}_\iota, d_{\widetilde{\iota}})$, according to (23) yields

$$V_{i\tilde{\sigma}_\iota}(d_{\widetilde{\iota}}^-) \leq V_{i\tilde{\sigma}_\iota}(\tilde{d}_\iota^+), k \in [\tilde{d}_\iota, d_{\widetilde{\iota}}) \tag{38}$$

For controller switching instant $\tilde{d}_\iota$ caused by deception attacks, we get

$$V_{i\tilde{\sigma}_\iota}(\tilde{d}_\iota^+) = V_{ij}(\tilde{d}_\iota^-) + X^T(\tilde{d}_\iota)N_{ij,i\tilde{\sigma}_\iota}X(\tilde{d}_\iota) \tag{39}$$

On the interval $[\kappa_q, \tilde{d}_\iota)$, according to (23) yields

$$V_{ij}(\tilde{d}_\iota^-) \leq V_{ij}(\kappa_q^+), k \in [\kappa_q, \tilde{d}_\iota) \tag{40}$$

Combining (14)-(15) and (37)-(40), it can be deduced that

$$
\begin{aligned}
V_{ii}(\kappa_{q+1}^-) &\leq V_{ii}(d_{\widetilde{\iota}}^+) = V_{i\tilde{\sigma}_\iota}(d_{\widetilde{\iota}}^-) + X^T(d_{\widetilde{\iota}})N_{i\tilde{\sigma}_\iota,i}X(d_{\widetilde{\iota}}) \\
&\leq V_{i\tilde{\sigma}_\iota}(\tilde{d}_\iota^+) + X^T(d_{\widetilde{\iota}})N_{i\tilde{\sigma}_\iota,i}X(d_{\widetilde{\iota}}) \\
&= V_{ij}(\tilde{d}_\iota^-) + X^T(\tilde{d}_\iota)N_{ij,i\tilde{\sigma}_\iota}X(\tilde{d}_\iota) + X^T(d_{\widetilde{\iota}})N_{i\tilde{\sigma}_\iota,i}X(d_{\widetilde{\iota}}) \\
&\leq V_{ij}(\kappa_q^+) + X^T(\tilde{d}_\iota)N_{ij,i\tilde{\sigma}_\iota}X(\tilde{d}_\iota) + X^T(d_{\widetilde{\iota}})N_{i\tilde{\sigma}_\iota,i}X(d_{\widetilde{\iota}}) \\
&\leq V_{ij}(\kappa_q^+) + X^T(\tilde{d}_\iota)(N_{ij,i\tilde{\sigma}_\iota} + N_{i\tilde{\sigma}_\iota,i})X(\tilde{d}_\iota) \\
&\leq V_{ij}(\kappa_q^+) + X^T(\kappa_q)N_{ij,i}X(\kappa_q)
\end{aligned} \tag{41}
$$

**Case D.** On the interval $[\kappa_q, \tilde{d}_\iota)$, $(\sigma, \phi) = (i, j)$. On the interval $[\tilde{d}_\iota, d_{\widetilde{\iota}})$, $(\sigma, \phi) = (i, \tilde{\sigma}_\iota)$. On the interval $[d_{\widetilde{\iota}}, \tilde{d}_{\bar{\iota}+1})$, $(\sigma, \phi) = (i, i)$, $\bar{\iota} \in \{\iota, \iota+1, \cdots, \iota+\iota_q\}$, $\tilde{d}_{\iota+\iota_q+1} = \kappa_{q+1}$. Analyses on interval $[\tilde{d}_{\iota+1}, \kappa_{q+1}]$ are the same as in case B.

On the interval $[\tilde{d}_\iota, \kappa_{q+1})$, combining (33)-(35) and (38)-(39) yields

$$
\begin{aligned}
V_{ii}(\kappa_{q+1}^-) &\leq V_{ii}(\tilde{d}_{\iota+\iota_q}^-) \leq \cdots \leq V_{ii}(\tilde{d}_{\iota+1}^-) \leq V_{ii}(d_{\widetilde{\iota}}^+) \\
&= V_{i\tilde{\sigma}_\iota}(d_{\widetilde{\iota}}^-) + X^T(d_{\widetilde{\iota}})N_{i\tilde{\sigma}_\iota,i}X(d_{\widetilde{\iota}}) \\
&\leq V_{i\tilde{\sigma}_\iota}(\tilde{d}_\iota^+) + X^T(d_{\widetilde{\iota}})N_{i\tilde{\sigma}_\iota,i}X(d_{\widetilde{\iota}}) \\
&= V_{ij}(\tilde{d}_\iota^-) + X^T(\tilde{d}_\iota)N_{ij,i\tilde{\sigma}_\iota}X(\tilde{d}_\iota) + X^T(d_{\widetilde{\iota}})N_{i\tilde{\sigma}_\iota,i}X(d_{\widetilde{\iota}})
\end{aligned} \tag{42}
$$

On the interval $[\kappa_q, \tilde{d}_\iota)$, according to (23) yields

$$V_{ij}(\tilde{d}_\iota^-) \leq V_{ij}(\kappa_q^+), k \in [\kappa_q, \tilde{d}_\iota) \tag{43}$$

Combining (14)-(15) and (42)-(43), it can be deduced that

$$
\begin{aligned}
V_{ii}(\kappa_{q+1}^-) &\le V_{ij}(\tilde{d}_\iota^-) + X^T(\tilde{d}_\iota)N_{ij,i\tilde{\sigma}_\iota}X(\tilde{d}_\iota) + X^T(d_{\sim\iota})N_{i\tilde{\sigma}_\iota,i}X(d_{\sim\iota}) \\
&\le V_{ij}(\kappa_q^+) + X^T(\tilde{d}_\iota)(N_{ij,i\tilde{\sigma}_\iota} + N_{i\tilde{\sigma}_\iota,i})X(\tilde{d}_\iota) \\
&\le V_{ij}(\kappa_q^+) + X^T(\kappa_q)N_{ij,i}X(\kappa_q)
\end{aligned}
\tag{44}
$$

**Case E.** On the interval $[\kappa_q, \tilde{\kappa}_q)$, $(\sigma, \phi) = (i, j)$. On the interval $[\tilde{\kappa}_q, \tilde{d}_\iota)$, $(\sigma, \phi) = (i, i)$. On the interval $[\tilde{d}_{\bar{\iota}}, d_{\sim\bar{\iota}})$, $(\sigma, \phi) = (i, \tilde{\sigma}_{\bar{\iota}})$. On the interval $[d_{\sim\bar{\iota}}, \tilde{d}_{\bar{\iota}+1})$, $(\sigma, \phi) = (i, i)$, $\bar{\iota} \in \{\iota, \iota+1, \cdots, \iota+\iota_q\}$, $\tilde{d}_{\iota+\iota_q+1} = \kappa_{q+1}$. Analyses on interval $[\tilde{d}_\iota, \kappa_{q+1})$ are the same as in case D, we get

$$
\begin{aligned}
V_{ii}(\kappa_{q+1}^-) &\le V_{ii}(\tilde{d}_{\iota+\iota_q}^-) \le \cdots \le V_{ii}(\tilde{d}_{\iota+1}^-) \le V_{ii}(d_{\sim\iota}^+) \\
&= V_{i\tilde{\sigma}_\iota}(d_{\sim\iota}^-) + X^T(d_{\sim\iota})N_{i\tilde{\sigma}_\iota,i}X(d_{\sim\iota}) \\
&\le V_{i\tilde{\sigma}_\iota}(\tilde{d}_\iota^+) + X^T(d_{\sim\iota})N_{i\tilde{\sigma}_\iota,i}X(d_{\sim\iota}) \\
&= V_{ii}(\tilde{d}_\iota^-) + X^T(\tilde{d}_\iota)N_{i,i\tilde{\sigma}_\iota}X(\tilde{d}_\iota) + X^T(d_{\sim\iota})N_{i\tilde{\sigma}_\iota,i}X(d_{\sim\iota})
\end{aligned}
\tag{45}
$$

On the interval $[\tilde{\kappa}_q, \tilde{d}_\iota)$, according to (23) yields

$$
V_{ii}(\tilde{d}_\iota^-) \le V_{ii}(\tilde{\kappa}_q^+), k \in [\tilde{\kappa}_q, \tilde{d}_\iota)
\tag{46}
$$

For controller switching instant $\tilde{\kappa}_q$ affected by sub-system switching instant, according to (4) yields

$$
V_{ii}(\tilde{\kappa}_q^+) = V_{ij}(\tilde{\kappa}_q^-) + X^T(\tilde{\kappa}_q)N_{ij,i}X(\tilde{\kappa}_q)
\tag{47}
$$

On the interval $[\kappa_q, \tilde{\kappa}_q)$, according to (23) yields

$$
V_{ij}(\tilde{\kappa}_q^-) \le V_{ij}(\kappa_q^+), k \in [\kappa_q, \tilde{\kappa}_q)
\tag{48}
$$

Combining (14)-(15) and (45)-(48), it can be deduced that

$$
\begin{aligned}
V_{ii}(\kappa_{q+1}^-) &\le V_{ii}(\tilde{d}_\iota^-) + X^T(\tilde{d}_\iota)N_{i,i\tilde{\sigma}_\iota}X(\tilde{d}_\iota) + X^T(d_{\sim\iota})N_{i\tilde{\sigma}_\iota,i}X(d_{\sim\iota}) \\
&\le V_{ii}(\tilde{\kappa}_q^+) + X^T(\tilde{d}_\iota)N_{i,i\tilde{\sigma}_\iota}X(\tilde{d}_\iota) + X^T(d_{\sim\iota})N_{i\tilde{\sigma}_\iota,i}X(d_{\sim\iota}) \\
&= V_{ij}(\tilde{\kappa}_q^-) + X^T(\tilde{\kappa}_q)N_{ij,i}X(\tilde{\kappa}_q) + X^T(\tilde{d}_\iota)N_{i,i\tilde{\sigma}_\iota}X(\tilde{d}_\iota) \\
&\quad + X^T(d_{\sim\iota})N_{i\tilde{\sigma}_\iota,i}X(d_{\sim\iota}) \\
&\le V_{ij}(\kappa_q^+) + X^T(\tilde{\kappa}_q)N_{ij,i}X(\tilde{\kappa}_q) \\
&\quad + X^T(\tilde{d}_\iota)(N_{i,i\tilde{\sigma}_\iota} + N_{i\tilde{\sigma}_\iota,i})X(\tilde{d}_\iota) \\
&\le V_{ij}(\kappa_q^+) + X^T(\kappa_q)N_{ij,i}X(\kappa_q)
\end{aligned}
\tag{49}
$$

**Case F.** On the interval $[\kappa_q, \tilde{\kappa}_q)$, $(\sigma, \phi) = (i, j)$. On the interval $[\tilde{\kappa}_q, \kappa_{q+1})$, $(\sigma, \phi) = (i, i)$. According to (23) yields

$$
V_{ii}(\kappa_{q+1}^-) \le V_{ii}(\tilde{\kappa}_q^+), k \in [\tilde{\kappa}_q, \kappa_{q+1})
\tag{50}
$$

Combining (14)-(15) and (48)-(50), it can be deduced that

$$
\begin{aligned}
V_{ii}(\kappa_{q+1}^-) &\le V_{ii}(\tilde{\kappa}_q^+) = V_{ij}(\tilde{\kappa}_q^-) + X^T(\tilde{\kappa}_q)N_{ij,i}X(\tilde{\kappa}_q) \\
&\le V_{ij}(\kappa_q^+) + X^T(\kappa_q)N_{ij,i}X(\kappa_q)
\end{aligned}
\tag{51}
$$

**Case G.** On the interval $[\kappa_q, \tilde{\kappa}_q)$, $(\sigma, \phi) = (i, j)$. On the interval $[\tilde{\kappa}_q, \tilde{d}_\iota)$, $(\sigma, \phi) = (i, i)$. On the interval $[\tilde{d}_\iota, \kappa_{q+1})$, $(\sigma, \phi) = (i, \tilde{\sigma}_\iota)$. According to (23) yields

$$
V_{i\tilde{\sigma}_\iota}(\kappa_{q+1}^-) \le V_{i\tilde{\sigma}_\iota}(\tilde{d}_\iota^+), k \in [\tilde{d}_\iota, \kappa_{q+1})
\tag{52}
$$

For controller switching instant $\tilde{d}_\iota$ caused by deception attacks, we get

$$
V_{i\tilde{\sigma}_\iota}(\tilde{d}_\iota^+) = V_{ii}(\tilde{d}_\iota^-) + X^T(\tilde{d}_\iota)N_{i,i\tilde{\sigma}_\iota}X(\tilde{d}_\iota)
\tag{53}
$$

Analyses on interval $[\kappa_q, \tilde{d}_\iota)$ are the same as in case E. Combining (14)-(15) and (46)-(48) and (52)-(53), it can be deduced that

$$
\begin{aligned}
V_{i\tilde{\sigma}_\iota}(\kappa_{q+1}^-) &\le V_{i\tilde{\sigma}_\iota}(\tilde{d}_\iota^+) = V_{ii}(\tilde{d}_\iota^-) + X^T(\tilde{d}_\iota)N_{i,i\tilde{\sigma}_\iota}X(\tilde{d}_\iota) \\
&\le V_{ii}(\tilde{\kappa}_q^+) + X^T(\tilde{d}_\iota)N_{i,i\tilde{\sigma}_\iota}X(\tilde{d}_\iota) \\
&= V_{ij}(\tilde{\kappa}_q^-) + X^T(\tilde{\kappa}_q)N_{ij,i}X(\tilde{\kappa}_q) + X^T(\tilde{d}_\iota)N_{i,i\tilde{\sigma}_\iota}X(\tilde{d}_\iota) \\
&\le V_{ij}(\kappa_q^+) + X^T(\tilde{\kappa}_q)(N_{ij,i} + N_{i,i\tilde{\sigma}_\iota})X(\tilde{\kappa}_q) \\
&\le V_{ij}(\kappa_q^+) + X^T(\kappa_q)N_{ij,i\tilde{\sigma}_\iota}X(\kappa_q)
\end{aligned}
\tag{54}
$$

Thus, for the changes (27), (36), (41), (44), (49), (51) and (54) of (19) in any interval $[\kappa_q, \kappa_{q+1}]$ under Cases A-G, it can be summarised through conditions $N_{i\tilde{\phi}_1,i\tilde{\phi}_2} + N_{i\tilde{\phi}_2,i\tilde{\phi}_1} \le N_{i\tilde{\phi}_1,i\tilde{\phi}_1}$ and $N_{i,i} = 0$ as

$$
V_{i\ell}(\kappa_{q+1}^-) \le V_{i\phi}(\kappa_q^+) + X^T(\kappa_q)N_{i\phi,i\ell}X(\kappa_q)
\tag{55}
$$

with $[i\phi, i\ell, case] \in \{(i\tilde{\sigma}_\iota, i, A\text{-}B), (ij, i, C\text{-}F), (ij, i\tilde{\sigma}_\iota, G)\}$.

Thirdly, discussing the change of (19) after the subsystem switching instant $\kappa_q$, for the subsystem switching instant $\kappa_q$ under the optimal prediction-based switching rule (10) as

$$
V_{i\phi}(\kappa_q^+) = V_{j\phi}(\kappa_q^-) + X^T(\kappa_q)N_{j\phi,i\phi}X(\kappa_q), \phi \in \{\tilde{\sigma}_\iota, j\}
\tag{56}
$$

Fourthly, discuss the relationship between $V(k)$ and $V(\kappa_0)$ over the entire running interval $[0, k)$. From conditions (15) and (55)-(56) it can be deduced that

$$
\begin{aligned}
V_{i\ell}(\kappa_{q+1}^-) &\le V_{i\phi}(\kappa_q^+) + X^T(\kappa_q)N_{i\phi,i\ell}X(\kappa_q) \\
&= V_{j\phi}(\kappa_q^-) + X^T(\kappa_q)N_{j\phi,i\phi}X(\kappa_q) + X^T(\kappa_q)N_{i\phi,i\ell}X(\kappa_q) \\
&= V_{j\phi}(\kappa_q^-) + X^T(\kappa_q)(N_{j\phi,i\phi} + N_{i\phi,i\ell})X(\kappa_q) \\
&= V_{j\phi}(\kappa_q^-) + X^T(\kappa_q)N_{j\phi,i\ell}X(\kappa_q) \\
&\le V_{\vartheta(\kappa_{q-1}^-)}(\kappa_{q-1}^-) + X^T(\kappa_{q-1})N_{\vartheta(\kappa_{q-1}^-),j\phi}X(\kappa_{q-1}) \\
&\quad + X^T(\kappa_q)N_{j\phi,i\ell}X(\kappa_q) \\
&\le V_{\vartheta(\kappa_{q-1}^-)}(\kappa_{q-1}^-) + X^T(\kappa_{q-1})N_{\vartheta(\kappa_{q-1}^-),i\ell}X(\kappa_{q-1}) \le \cdots \\
&\le V_{\vartheta(\kappa_1^-)}(\kappa_1^-) + X^T(\kappa_1)N_{\vartheta(\kappa_1^-),i\ell}X(\kappa_1) \\
&\le V_{\vartheta(\kappa_0)}(\kappa_0)
\end{aligned}
\tag{57}
$$

From (23), we have $\Delta V_\vartheta(k+l|k) + J_\infty(k+l|k) < 0$. Adding its two sides from $l = 0$ to $l = \infty$ yields

$$
\begin{aligned}
\mathcal{G}_\infty = \sum_{l=0}^{\infty} J_\infty(k+l|k) &< -\sum_{l=0}^{\infty} \Delta V_\vartheta(k+l|k) \\
&= V_\vartheta(k|k) - V_\vartheta(\infty) < V_\vartheta(k|k)
\end{aligned}
\tag{58}
$$

According to condition (17), it implies that the upper bound of the quadratic cost function $\mathcal{G}_\infty$ is $\beta$. Up to this point, the asymptotic stability of the closed-loop system (11) is proved and the control objective (i) is achieved.

To complete the proof of control objective (ii). Firstly, for the control input constraint $\mathbb{U}$, the invariant ellipsoid is constructed as

$$\mathbb{S} = \{X| \; \|X^T P_\vartheta X\| \leq \beta\} \qquad (59)$$

implies that

$$
\begin{aligned}
\max_k\{|\hat{u}(k)|^2\} &= \max_k\{|K_{\tilde{\sigma}}\hat{x}(k)|^2\}\\
&= \max_k\{|K_{\tilde{\sigma}}\hat{I}X(k)|^2\}\\
&= \max_k\{|K_{\tilde{\sigma}}\hat{I}P_\vartheta^{-\frac{1}{2}}|^2\}\{|P_\vartheta^{\frac{1}{2}}X(k)|^2\} \qquad (60)\\
&\leq \beta|K_{\tilde{\sigma}}\hat{I}P_\vartheta^{-\frac{1}{2}}|^2\\
&= \beta K_{\tilde{\sigma}}\hat{I}P_\vartheta^{-1}\hat{I}^T K_{\tilde{\sigma}}^T
\end{aligned}
$$

At this point, the control input constraints are proved and the control objective (ii) is achieved.

Next, we handle nonlinear terms in (12) and (14).

**Theorem 2:** Given positive scalars $\bar{\eta}, \eta, \gamma_\sigma < 1$, model prediction control weight matrix $S_{\hat{x}} > 0$, $S_{\hat{u}} > 0$ and $\varsigma_{\tilde{\vartheta},\bar{\vartheta}} < 0$. There exist matrices $P_\vartheta > 0$, $Q_\vartheta > 0$, $R_\vartheta > 0$, $\Phi_\sigma^1 > 0$, $\Phi_\sigma^2 > 0$ and $M_\vartheta^1$ with $(\vartheta, \sigma, \phi) \in \{(ii, i, i), (ij, i, j), (i\tilde{\sigma}_{\bar{\iota}}, i, \tilde{\sigma}_{\bar{\iota}})\}$, $\bar{\iota} \in \{\iota, \iota+1, \cdots, \iota+\iota_q\}$, scalar $\beta > 0$ and indeterminate matrix $N_{\tilde{\vartheta},\bar{\vartheta}}$ satisfying

$$\bar{\Psi}_\vartheta < 0 \qquad (61)$$

$$\bar{\Omega}_\vartheta < 0 \qquad (62)$$

Then, $\beta$ is an upper bound for $\mathcal{G}_\infty(k)$ and the closed-loop system (11) is asymptotically stable. The matrix $\bar{\Psi}_\vartheta = \{\bar{\varphi}_\vartheta^{\bar{\varrho}_1,\bar{\varrho}_2}\}, (\bar{\varrho}_1, \bar{\varrho}_2 = 1, \cdots, 11)$ is composed of sub-blocks:

$$
\begin{aligned}
\bar{\varphi}_\vartheta^{1,1} &= -P_\vartheta + \tilde{R}_\vartheta + Q_\vartheta^0 - R_\vartheta^0 + \hat{I}^T S_{\hat{x}}\hat{I} + \hat{I}^T K_\phi^T S_{\hat{u}} K_\phi \hat{I} + N_{\tilde{\vartheta},\bar{\vartheta}}\\
&\quad + \sum_{\upsilon=1}^{\bar{p}} \varsigma_{\tilde{\vartheta},\bar{\vartheta}}(P_{\tilde{\vartheta}} - P_\vartheta + N_{\tilde{\vartheta},\bar{\vartheta}}), \bar{\varphi}_\vartheta^{1,3} = R_\vartheta^0, \bar{\varphi}_\vartheta^{1,7} = (\Lambda_\vartheta^1)^T,
\end{aligned}
$$

$$\bar{\varphi}_\vartheta^{1,11} = \tilde{R}_\vartheta^T, \bar{\varphi}_\vartheta^{2,2} = -2R_\vartheta^1 + M_\vartheta^1 + (M_\vartheta^1)^T + \gamma_\sigma Z^T \Phi_\sigma^2 Z,$$

$$\bar{\varphi}_\vartheta^{2,3} = R_\vartheta^1 - (M_\vartheta^1)^T, \bar{\varphi}_\vartheta^{2,4} = R_\vartheta^1 - M_\vartheta^1, \bar{\varphi}_\vartheta^{2,8} = (\Lambda_\vartheta^2)^T,$$

$$\bar{\varphi}_\vartheta^{3,3} = -Q_\vartheta^0 + Q_\vartheta^1 - R_\vartheta^0 - R_\vartheta^1, \bar{\varphi}_\vartheta^{3,4} = M_\vartheta^1, \bar{\varphi}_\vartheta^{4,4} = -Q_\vartheta^1 - R_\vartheta^1,$$

$$\bar{\varphi}_\vartheta^{5,5} = -\Phi_\sigma^1, \bar{\varphi}_\vartheta^{5,9} = (\Lambda_\vartheta^3)^T, \bar{\varphi}_\vartheta^{6,10} = (\Lambda_\vartheta^4)^T,$$

$$\bar{\varphi}_\vartheta^{7,7} = \bar{\varphi}_\vartheta^{8,8} = \bar{\varphi}_\vartheta^{9,9} = \bar{\varphi}_\vartheta^{10,10} = \lambda^2[4(P_\vartheta + \tilde{R}_\vartheta) - \tilde{R}_\vartheta] - 2\lambda I,$$

$$\bar{\varphi}_\vartheta^{11,11} = 4\lambda^2 \tilde{R}_\vartheta - 2\lambda I, \tilde{R}_\vartheta = \eta^2 R_\vartheta^0 + (\bar{\eta} - \eta)^2 R_\vartheta^1, \hat{I} = \begin{bmatrix} 1 & 0 \end{bmatrix},$$

$$Z = \begin{bmatrix} 1 & 1 \end{bmatrix}, (\vartheta, \sigma, \phi) \in \{(i, i, i), (ij, i, j), (i\tilde{\sigma}_{\bar{\iota}}, i, \tilde{\sigma}_{\bar{\iota}})\},$$

$$\upsilon \in \{i, j, \tilde{\sigma}_{\bar{\iota}}\}, (\tilde{\vartheta}, \bar{\vartheta}) \in \{(j\tilde{\sigma}_\iota, i\tilde{\sigma}_\iota), (j, ij), (i\tilde{\sigma}_{\bar{\iota}}, i), (i, i\tilde{\sigma}_{\bar{\iota}}),$$

$$(ij, i\tilde{\sigma}_\iota), (ij, i)\}, (\tilde{\vartheta}_1, \bar{\vartheta}_1) \subseteq (\tilde{\vartheta}, \bar{\vartheta}), (\tilde{\vartheta}_1, \bar{\vartheta}_2) \subseteq (\tilde{\vartheta}, \bar{\vartheta}),$$

$$(i\phi_1, i\phi_2) \in \{(i\tilde{\sigma}_{\bar{\iota}}, i), (i, i\tilde{\sigma}_{\bar{\iota}}), (ij, i\tilde{\sigma}_\iota), (ij, i)\} \subseteq (\tilde{\vartheta}, \bar{\vartheta}), (j\phi, i\phi)$$

$$\in \{(j\tilde{\sigma}_\iota, i\tilde{\sigma}_\iota), (j, ij)\} \subseteq (\tilde{\vartheta}, \bar{\vartheta})$$

$$P_{\tilde{\vartheta}} = \begin{bmatrix} \mathcal{P}_{\tilde{\vartheta}} & 0 \\ 0 & 0 \end{bmatrix}, P_{\bar{\vartheta}} = \begin{bmatrix} \mathcal{P}_{\bar{\vartheta}} & 0 \\ 0 & 0 \end{bmatrix}, N_{\tilde{\vartheta},\bar{\vartheta}} = \begin{bmatrix} \eta_{\tilde{\vartheta},\bar{\vartheta}} & 0 \\ 0 & 0 \end{bmatrix}.$$

The matrix $\bar{\Omega}_\vartheta = \{\bar{\zeta}_\vartheta^{\bar{\pi}_1,\bar{\pi}_2}\}, (\bar{\pi}_1, \bar{\pi}_2 = 1, \cdots, 4)$ is composed of sub-blocks:

$$\bar{\zeta}_\vartheta^{1,1} = -N_{\tilde{\vartheta},\bar{\vartheta}}, \bar{\zeta}_\vartheta^{1,5} = (\Lambda_\vartheta^1)^T, \bar{\zeta}_\vartheta^{2,6} = (\Lambda_\vartheta^2)^T, \bar{\zeta}_\vartheta^{3,7} = (\Lambda_\vartheta^3)^T,$$

$$\bar{\zeta}_\vartheta^{4,8} = (\Lambda_\vartheta^4)^T, \bar{\zeta}_\vartheta^{5,5} = \bar{\zeta}_\vartheta^{6,6} = \bar{\zeta}_\vartheta^{7,7} = \bar{\zeta}_\vartheta^{8,8} = 4\lambda^2 N_{\tilde{\vartheta},\bar{\vartheta}} - 2\lambda I.$$

The other block matrices in $\bar{\Psi}_\vartheta$ and $\bar{\Omega}_\vartheta$ are $\mathbf{0}$ in the appropriate dimensions.

**Proof:** Applying Lemma from [13] yields $-[4(P_\vartheta + \tilde{R}_\vartheta) - \tilde{R}_\vartheta]^{-1} < \lambda^2[4(P_\vartheta + \tilde{R}_\vartheta) - \tilde{R}_\vartheta] - 2\lambda I$ and $-4N_{\tilde{\vartheta},\bar{\vartheta}}^{-1} < 4\lambda^2 N_{\tilde{\vartheta},\bar{\vartheta}} - 2\lambda I$. Then, with the help of lemma [12], $\Psi_\vartheta < 0$ can be deduced from $\bar{\Psi}_\vartheta < 0$, and $\Omega_\vartheta < 0$ from $\bar{\Omega}_\vartheta < 0$. That is, (61)-(62) ensures that (12) and (14) hold. The proof is complete.

## V. SIMULATION EXAMPLE

In this section, the feasibility of the proposed contribution is verified by a NUMV example [1]. Select parameters $\eta = 0.02$, $\bar{\eta} = 0.2$, $\gamma_1^x = 0.2$, $\gamma_2^x = 0.1$ and model prediction control weight matrix $S_{\hat{x}} = 0.01$, $S_{\hat{u}} = 0.1$. The sampling period is 0.1 s. By solving conditions (61)-(62), the controller gain and event-triggering parameters can be calculated as

$$K_1 = \begin{bmatrix} -30 & 0 & 0 & -114.29 & 0 & 0 \\ 0 & -36.28 & -4.57 & 0 & -140.06 & 14.39 \\ 0 & -2.77 & -3.02 & 0 & 11 & -10 \end{bmatrix}$$

$$K_2 = \begin{bmatrix} -28.82 & 0 & 0 & -111.23 & 0 & 0 \\ 0 & -35.32 & 3.87 & 0 & -136 & 11.38 \\ 0 & 2.3 & -2.68 & 0 & 9.46 & -9.8 \end{bmatrix}$$

$$\Phi_1 = \begin{bmatrix} 5.8760 & 0 & 0 & -0.9585 & 0 & 0 \\ 0 & 5.8227 & 0.1481 & 0 & -0.0828 & -0.1597 \\ 0 & 0.1481 & 4.7006 & 0 & 0.7615 & -0.4513 \\ -0.9585 & 0 & 0 & 4.9126 & 0 & 0 \\ 0 & -0.0828 & 0.7615 & 0 & 3.7930 & 0.8140 \\ 0 & -0.1597 & -0.4513 & 0 & 0.8140 & 1.1503 \end{bmatrix}$$

$$\Phi_2 = \begin{bmatrix} 11.5239 & 0 & 0 & -2.0207 & 0 & 0 \\ 0 & 0.9918 & 0.0658 & 0 & 0.3070 & 0.1441 \\ 0 & 0.0658 & -0.1827 & 0 & -0.0135 & 0.0232 \\ -2.0207 & 0 & 0 & 9.7748 & 0 & 0 \\ 0 & 0.3070 & -0.0135 & 0 & 0.5952 & -0.1300 \\ 0 & 0.1441 & 0.0232 & 0 & -0.1300 & -0.1450 \end{bmatrix}$$

Fig. 2 illustrates the depicted deception attack. Under the designed optimal security control policy based on model prediction, the optimal state trajectories and control inputs are plotted in Fig. 3 and Fig. 7, respectively. Fig 4 illustrates the system state trajectory without optimal prediction parameter error detection, comparing with the optimal state trajectory in Fig. 3, it can be clearly seen that the event-triggering mechanism based on the optimal prediction can converge faster and smoother, and cope with the prediction deviation effectively. Fig. 5 and Fig.6 show the state trajectories and control inputs under observation-dependent state switching signals,

respectively. Fig. 8 and Fig. 9 compare the subsystem switching signals of the optimal state-dependent design with the subsystem switching signals of the observed state-dependent design. The subsystem switching in Fig. 8 occurs at 4.1s, 10.6s, 18.7s, 29.2s, 36.8s and 47s for a total of six switching behaviours. While the number of subsystem switching occurs more frequently in Fig. 9 for the subsystem switching signals designed by relying on the observed states, which shows that the optimal state-based switching rule designed in this paper effectively avoids unnecessary switching, especially when the system state fluctuates near the switching boundary. Combined with the comparison of state trajectories, the advantages of the predictive state-based switching rule are verified to help select the optimal subsystem during switching and optimise the overall performance of the system thereby improving the safety of the system.

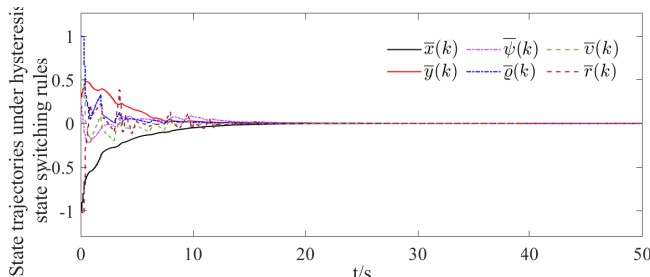

Fig. 5: State trajectories under hysteresis state switching rule .

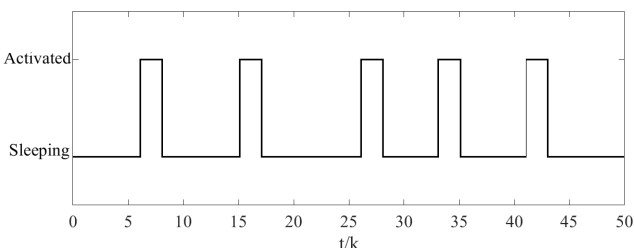

Fig. 2: Deception attacks .

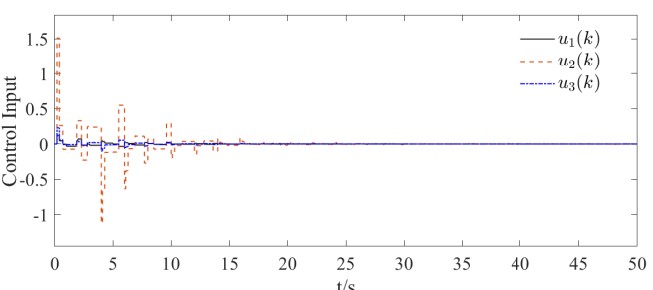

Fig. 6: System control inputs .

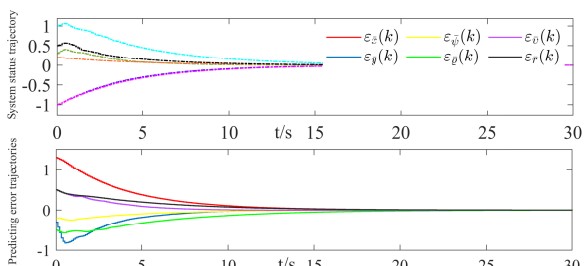

Fig. 3: Optimal state trajectory and error under optimal security control strategy .

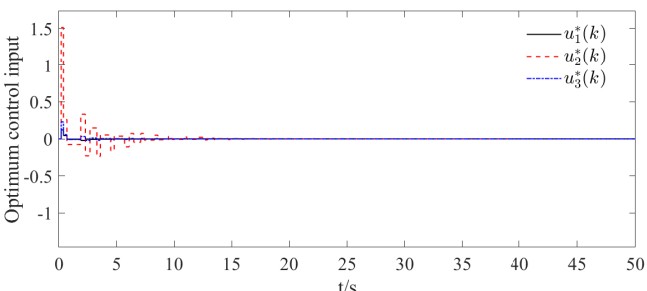

Fig. 7: Optimal system control inputs .

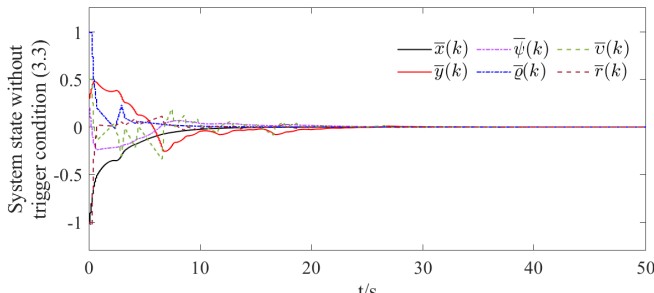

Fig. 4: The system state trajectories without optimal prediction parameter error detection condition .

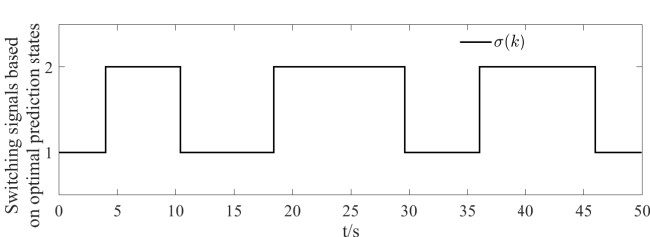

Fig. 8: The subsystem switching signal based on the optimal predicted states .

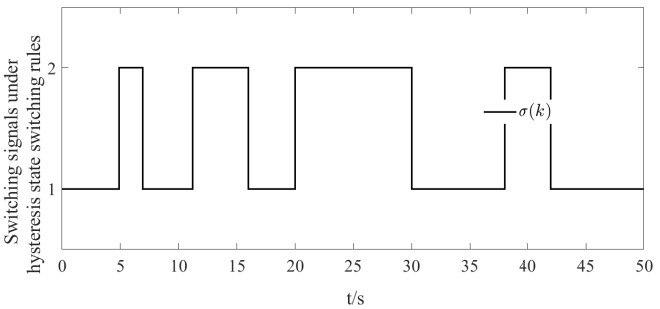

Fig. 9: Subsystem switching signal under hysteresis state switching rule .

## VI. CONCLUSION

In this paper, we study the security control problem of discrete switching system based on model prediction under spoofing attack. An event triggering mechanism based on optimal prediction is designed to shorten the asynchronous time between subsystems and controllers, which helps the system to make timely adjustments when the prediction deviation occurs, and reduces the impact of the attack on the performance of the system; a new quadratic cost function is proposed to take into account the extra cost of modal switching of the controllers in solving the optimal security control policy, and to avoid unnecessary and frequent switching, and at the same time, find a smoother and more stable control policy. A smoother and more stable control strategy is found to reduce the system fluctuation caused by switching; a subsystem switching rule based on the optimal prediction state is designed to select the optimal subsystem switching to optimise the overall system performance; sufficient conditions for asymptotic stability of the closed-loop system under the above event-triggering mechanism and optimal safety control strategy are given, and the validity of the proposed method is verified by the unmanned ship mass-switching dynamic positioning system model.

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
