# OpenReview forum: "Model-based Predictive Security Control for Discrete Switching Systems Under Deception Attacks"
_IEEE.org/ICIST/2024/Conference — IEEE ICIST 2024 Conference Submission_

### Official Review · Reviewer_HkWy · 2024-08-21
**accept**

**Rating:** 8
**Confidence:** 3

**Review:**

This paper considered the model-based predictive control for discrete switching systems under deception attacks. The theory is correct and can be accepted after responding the following comments.
(1)	In the introduction, it is not enough to state the current work. It should be expended and reconstructed.
(2)	In the simulation section, more analysis can be added to better explain the main results of this paper, that's not enough.
(3)	There are many typos and grammar errors. The authors should have a native English speaker or software packages to perform the editing check.

---

### Official Review · Reviewer_GZHn · 2024-08-28
**The obtained result is valuable and can be accepted after responding the following comments.**

**Rating:** 8
**Confidence:** 4

**Review:**

This article focuses on the development of a model-based predictive security control strategy for discrete switching systems under deception attacks. This article presents a comprehensive framework for predictive security control of discrete switching systems under deception attacks. The proposed method not only improves the security and stability of the system but also provides theoretical insights and practical guidelines for the design of such systems.The obtained result is valuable and can be accepted after responding the following comments.
 (1) In the introduction, it is not enough to state the current work. It should be expended and reconstructed. (2) In the simulation section, more analysis can be added to better explain the main results of this paper, that's not enough. (3) There are many typos and grammar errors. The authors should have a native English speaker or software packages to perform the editing check. (4) The conclusion of the article suggests using the present perfect tense for description

---

### Official Review · Reviewer_YUW9 · 2024-08-30
**This paper focuses on model prediction-based security control for discrete switching systems under deception attacks. An event-triggering mechanism based on the optimal prediction is designed to shorten the asynchronous time between the subsystem and the controller, which helps the system adjust when the prediction deviates and reduces the impact of the attack on the system performance. The referee thinks it contains some publishable materials and it is worthy of publishing after some revisions.**

**Rating:** 7
**Confidence:** 3

**Review:**

1. What are the significant differences between this study and previous studies? The author needs more explicit emphasis.
2. The legend of Figure 9 should not overlap with the image, and the authors are requested to correct it.
3. The disadvantage of the proposed method and future work must be described in conclusion.

---

### Decision · Program_Chairs · 2024-09-06

Accept (Oral)